# Influence of Mutations of Conserved Arginines on Neuropeptide Binding in the DPP III Active Site

**DOI:** 10.3390/molecules28041976

**Published:** 2023-02-19

**Authors:** Antonija Tomić, Zrinka Karačić, Sanja Tomić

**Affiliations:** Division of Organic Chemistry and Biochemistry, Ruđer Bošković Institute, Bijenička Cesta 54, 10000 Zagreb, Croatia

**Keywords:** dipeptidyl peptides III, neuropeptides, enzyme activity, enzyme mutations, conserved residues, inhibitory assays

## Abstract

Dipeptidyl peptidase III (DPP III), a zinc exopeptidase, is involved in the final steps of intercellular protein degradation and has a marked affinity for opioid peptides such as enkephalins and endomorphins. Recently, we characterized a number of neuropeptides as potential substrates and inhibitors of human DPP III and provided an explanation for their differential behavior. These studies prompted us to investigate the influence of the conserved R399 and R669 on neuropeptides binding to DPP III. Measuring kinetic parameters in inhibitory assays, we found that mutation of R669 to Ala or Met significantly reduced the inhibitory properties of the slow substrates tynorphin and valorphin, whereas the effects on binding of the good substrates Arg_2_-2NA and Leu-enkephalin were small. Molecular dynamics simulations of wild-type (WT) and mutant DPP III complexes with Leu-enkephalin, tynorphin, valorphin, and Arg_2_-2NA in conjunction with calculations of binding free energies revealed that the lower inhibitory potency of slow substrates in the R669A mutant can be explained by the lower binding affinity of tynorphin and the higher propensity of valorphin to hydrolyze in the mutant than in WT. The R399A mutation was shown to affect the binding and/or hydrolysis of both good and slow substrates, with the effects on Leu-enkephalin being the most pronounced.

## 1. Introduction

Dipeptidyl peptidase III (DPP III, EC 3.4.14.4) is a monozinc metallopeptidase that hydrolyzes dipeptides from the N-terminus of its substrates consisting of three or more amino acids [1]. DPP III is the only member of the M49 metallopeptidase family and the only metalloenzyme among dipeptidyl peptidases. This two-domain exopeptidase is broadly distributed in human tissues, where it participates in the final stages of protein turnover. However, the pronounced affinity for some bioactive peptides (angiotensins and opioid peptides) suggests more specific functions (e.g., a role in blood pressure regulation and involvement in the mammalian pain regulatory system). In 2016, Cruz-Diaz et al. [2] found that DPP III degrades angiotensin (1–7) in human renal epithelial cells, and in a study by Pang and coworkers, DPP III was associated with the renin-angiotensin system (RAS) and demonstrated a therapeutic role of DPP III in lowering blood pressure in angiotensin II-infused hypertensive mice [3]. In the same year, we determined the reaction profile for human DPP III (hDPP III) catalyzed Leu-enkephalin hydrolysis with quantum mechanics–molecular mechanics (QM/MM) [4] calculations, and we showed that the reaction rate is determined by the attack of the activated water molecule on the carbonyl carbon of the second peptide bond (P1-P1’) and inversion of the nitrogen atom from the same peptide bond. The data on the high levels of DPP III in the superficial laminae of the dorsal horn of the rat spinal cord, where this enzyme is co-localized with enkephalins and endomorphins, and the finding that DPP III can degrade these opioid peptides in vitro support the theory of the role of DPP III in the mammalian endogenous pain regulatory system [5,6].

In addition, a number of studies were conducted highlighting the role of DPP III in regulating oxidative stress through its involvement in modifying the Nrf2-KEAP1 pathway and the importance of DPP III in cancer surveillance and development [7,8,9,10,11,12,13,14].

Despite the efforts of numerous scientists, knowledge about the role that DPP III might play in various physiological processes is still unclear. Interestingly, the available crystal structures of human DPP III are mostly complexes with natural peptides, mainly neuropeptides (PDB: 5EGY, 3T6B, 3T6J, 5E2Q, 5E33, 5E3A, 5E3C, and 5EHH) [15,16]. These structures provided better insight into the accommodation of natural peptides in the active site of DPP III and enabled a detailed QM/MM study that we recently performed for another neuropeptide, tynorphin [17], to explain the different behaviors of tynorphin and Leu-enkephalin. Unlike Leu-enkephalin, which is a good substrate for DPP III, tynorphin is considered a DPP III inhibitor or slow substrate. We have shown that both tynorphin and Leu-enkephalin are cleaved in the active site of DPP III via the same water-mediated mechanism with similar activation energies. However, while the steps following the cleavage of the peptide bond in the case of Leu-enkephalin favor the forward reaction, the energy barriers of the transition states in the case of tynorphin are such that the system can easily move between the final product and two of the preceding intermediates. We observed that the products of tynorphin hydrolysis are more tightly bound in the interdomain cleft of DPP III than in those of Leu-enkephalin. Therefore, the complete cycle of enzyme recovery in the case of tynorphin hydrolysis is slower than in the case of hydrolysis of Leu-enkephalin; thus, it could act as an inhibitor.

To identify potential new substrates of DPP III that would be a valuable aid in elucidating the role of DPP III in humans, we have investigated a number of neuropeptides both computationally and experimentally [18]. In the study, we identified valorphin, Leu-valorphin-Arg, β-casomorphin, and hemorphin-4 as in vitro substrates of hDPP III, and by combining several molecular dynamics (MD) simulation techniques, we found that good substrates such as Leu-enkephalin and hemorphin-4 are more suitable for hydrolysis than slow substrates such as valorphin and tynorphin. In this and in QM/MM studies of the DPP III enzyme mechanism, we found that the P3’ subsite (for the definition of peptide and enzyme subsites, see Appendix A) of neuropeptides with five or more amino acid residues is stabilized by R669. The C-terminus of tynorphin and Leu-enkephalin forms a salt bridge with R669, whereas valorphin interacts with R669 via hydrogen bond with carbonyl oxygen at the P3’ position. In addition, the indole group of tryptophan at the P3’ position of tynorphin and valorphin forms cation–π interactions with the guanidino group of R669, whereas the isobutyl group of leucine at the P3’ position of Leu-enkephalin is oriented in the opposite direction and forms CH–π interactions with F443 from the upper protein domain. Apparently, R669 plays an important role in the stabilization of peptide ligands. In contrast, Arg at position 399 is the only amino acid found to destabilize the binding of slow substrates, namely valorphin and tynorphin, while forming weak stabilizing interactions with tyrosine residues at position P2 of good substrates, namely Leu-enkephalin and hemorphin-4, and van der Waals interactions with the N-terminal Arg of Arg_2_-2NA [19].

R399, which is a part of the 2nd conserved region (G385-L409) of DPP III, is conserved in metazoa and some bacteria (those with the hexapeptide HEXXXH motif), whereas R669 is conserved only in metazoa [20]. To better understand the influence of these two conserved arginines located on opposite sides of the substrate binding site (Figure 1), R399 near the peptide N-terminus and R669 near the peptide C-terminus binding region (S2 and S3’ subsites, respectively), we performed additional experimental and computational studies on their mutants.

## 2. Results and Discussion

MD simulations 1 µs-long were performed for DPP III Arg mutants (R669A and R399A) in its unbound and ligand-bound states. The results were compared with the previously simulated wild-type enzyme, which was also simulated in its unbound and bound states [18]. According to the RMSD and R_g_ profiles shown in Appendix A, no significant change in protein structure (compared with an initially optimized protein structure) was observed during MD simulations for either the ligand-free DPP III or its complexes with the good DPP III substrates (Arg_2_-2NA and Leu-enkephalin). Apparently, these structures seem to be largely stable during the production phase, as only the flexibility of the loop comprising residues 463–489 increased in the ligand-free R669A mutant. Interestingly, DPP III interacted via this loop with KEAP1, the major oxidative stress sensor in humans. In contrast to the complexes with good substrates, the mutant DPP III complexes with the slow substrates, valorphin and especially tynorphin, showed a larger increase in RMSD values during the MD simulations compared to the initial structure, indicating the greatest changes in enzyme structure at the beginning of the production phase in the case of the DPP III–tynorphin complexes and during the first 300 or 600 ns in the case of the DPP III–valorphin complexes. The same behavior was observed for the WT DPP III–valorphin complex, whereas the structure of the WT DPP III–tynorphin complex was stable throughout the production phase. Although slightly larger R_g_ values were determined for the complexes with tynorphin and valorphin than for the complexes with Arg_2_-2NA and Leu-enkephalin or with unbound enzymes, the compact enzyme–ligand structure was preserved in all simulations.

### 2.1. Impact of R669A Mutation on Ligand Binding

R669 is a conserved amino acid residue within the DPP III family that is located in the loop (unstructured region) connecting the short β-strand (L671-V673) of the lower domain and the α-helix (S654-R665) of the upper domain. It is also covalently bound to K670, a residue that contributes to the hinge-like movement of the enzyme, according to Bezerra et al. [15].

As expected, the effects of the R669A mutation on the stabilization of the protein–peptide interaction were the most pronounced in the regions near the C-terminus of the peptide. Although the change in stabilization of the ligand residues at the P2’ and P3’ positions after mutation was observed in all complexes with peptides (Figure 2), it was the most pronounced in the complexes with the slow substrates (valorphin and tynorphin) with significantly better stabilization of the ligand residue (proline) at the P2’ position, whereas the opposite was true for the peptide residue (tryptophan) at the P3’ position (Figure 2). As expected, the less favorable binding of the P3’ residue of slow substrates in complex with the mutant enzyme was a consequence of the absence of hydrogen bond/salt bridge interactions (Figure 3 and Appendix A) as well as the cation–π interactions with R669 present in the WT complex. In the mutated complex with valorphin, they were partially compensated by the formation of a stable hydrogen bond between the backbone of the P3′ residue and the carbonyl group of I386 from the lower domain (Figure 3 and Figure 4), whereas in the complex with tynorphin, there were strong electrostatic interactions with R572 from the upper domain (Figure 4). The indole group of the tryptophan at P3’ forms not only CH–π interactions with the hinge region in the mutant complexes but also CH–π interactions with the side chain of I386 in the complex with valorphin, CH–π interactions with upper protein domain residues (such as F443), and occasionally cation–π interactions with the conserved K670 in the complex with tynorphin. The movement of tynorphin toward the upper protein domain observed in the complex with the R669 mutant was also assisted by the formation of a strong hydrogen bond between the carbonyl group of the proline at the P2’ position and R572 (Figure 3 and Figure 4). In the R669A–valorphin complex, the proline residue at the P2’ position was sufficiently distant that it could not form hydrogen bonds with R572; instead, it formed van der Waals interactions with the side chains of I386, H568, and the valine residue at the P1 position. Consequently, the stabilization of P2′ subsite in the mutant complex with valorphin was more pronounced than with tynorphin (Figure 2). The tighter binding of the P2’ and P3’ residues of valorphin than of tynorphin in the complex with mutant R669A compared with the complex with the enzyme WT is reflected in their lower RMSF values (Appendix A). In complexes with Leu-enkephalin, a slight decrease in the binding affinities of the P2’ and P3’ substrate residues was observed in the R669A mutant compared with the WT enzyme. The salt bridge interaction between the C-terminus of Leu-enkephalin and R669 in the WT complex was replaced by the salt bridge with R572 in the complex with the mutant protein upon rotation of the C-terminus of the peptide (see Figure 3). Thus, while in the complex with WT, the enzyme residue R572 formed only a hydrogen bond with the Leu-enkephalin residue at the P2’ position; in the complex R669A–Leu-enkephalin, it also formed a hydrogen bond with the peptide residue at the P3’ position. The rotation of the C-terminus of the peptide was also accompanied by a rotation of the corresponding amino acid side chain, which was then more oriented toward the lower domain (see Appendix A). Moreover, the cation–π interactions that the phenylalanine residue at the P2’ formed with R669 in the WT enzyme complex were replaced by the van der Waals interactions with M569 and H568 in the complex with the R669A mutant (the increased flexibility of the side chain of the P2’ phenylalanine residue gained by the mutation is shown in Appendix A).

The MD simulation results suggest that the R669A mutation also affects binding of the three amino acids from the peptide N-terminus. Namely, the R669A mutation impairs antiparallel binding of peptides to the β-strand from the lower protein domain, with the binding of tynorphin being most affected. This is evident from the elongation of the distances between the backbone of the tynorphin residue P2 and the E316 carboxyl group and that of the peptide residue P1’ and protein residues G389, A388, and E512 (Appendix A). As a result, the hydrogen bonding between these residues decreased (Figure 3). Moreover, the mutation resulted in the loss of hydrogen bonds between the tynorphin P1 residue and the highly conserved residues H568 and Y318, which are important for stabilizing the substrate during hydrolysis (Figure 3 and Appendix A) [4,17,21,22]. The less favorable binding of the P2, P1, and P1′ subsites of tynorphin to the R669A mutant compared to the WT enzyme was also confirmed via the MM/GBSA analysis (Figure 2).

In contrast to the effects of the R669A mutation on the binding of tynorphin, the binding of Leu-enkephalin and valorphin to the mutant in the form of a β-strand was only slightly affected. In the complexes with the mutant enzyme, we observed only an increase in the distance between the carbonyl group of A388 and the backbone of the peptide residue P1’ in the simulations of the complex with Leu-enkephalin and an increase in the distance between backbone of the residues N391 and N394 and the backbone of the residue in the P2 position of the peptide (Appendix A) in the simulations of the complex with valorphin. Consequently, the corresponding hydrogen bonds between proteins and ligands were weakened or lost (Figure 3). Interestingly, in the MD simulations of the R669A–valorphin complex, we observed stronger stabilizing interactions with residues E316, Y318, and H568 compared to the complex with the WT enzyme. This is evident from the shorter distances between the E316 carboxyl group and the backbone of peptide residue P2 as well as those between the side chain atoms of residues H568 and Y318 and the backbone atoms of peptide residue P1 (Appendix A) in the complex with the R669A mutant compared to the WT enzyme. In summary, the results of the MM/GBSA per-residue decomposition analysis indicated mostly unchanged stabilization of the first three amino acids at the Leu-enkephalin N-terminus and significantly more favorable binding for the valorphin valine residue at the P1 position in complexes with the R669A enzyme compared to the WT enzyme.

It should be mentioned that the strong Nε-H∙∙∙O-hydrogen bond between the Nε atom of protein residue H568 and the carbonyl oxygen atom of valine at the P1 position was present throughout the MD simulation of the R669A–valorphin complex, whereas in the simulations of the WT DPP III–valorphin complex, it was replaced after 300 ns by a weaker Cε-H∙∙∙O hydrogen bond (Figure 4a and Appendix A). This occurred in WT as a result of multiple bond restructuring caused by the formation of a strong Nε-H∙∙∙O-hydrogen bond between H568 and the carboxyl group E508, which was made possible by the rotation of the E508 residue after the breaking of the Y318–E508 hydrogen bond.

The experimental measurements performed in this work indicate that R669 plays a more important role in binding the slow peptide substrates valorphin and tynorphin (~30-fold increase in *K*_i_ value) than the good substrate Leu-enkephalin (~5-fold increase in *K*_i_ value) in the active site of DPP III (Table 1).

According to the MM/PBSA binding energies (Table 2), the R669A mutation had the most significant effect on the binding affinity of tynorphin. The large difference in the binding energy of tynorphin with the enzyme WT and the mutant R669A could explain the 30-fold increase in the experimentally determined inhibition constant (*K*_i_). Only a small difference in binding energy was determined for the complexes with Leu-enkephalin, indicating a slight preference for binding with an unmodified enzyme. This is consistent with the 5-fold higher *K*_i_ value experimentally observed for the binding of Leu-enkephalin with the R669A mutant compared to WT. In the case of valorphin, a slightly higher preference for binding to the mutant enzyme R669A compared to WT was indicated in our MM/PBSA energy calculations. The reason for the 30-fold increase of the inhibition constant by mutation should therefore be sought elsewhere. For example, MD simulations of the R669A–valorphin complex showed much more stable hydrogen bonding between the ligand and catalytically important protein residues Y318 and H568. Therefore, we could hypothesize that the reason for the lower inhibition of the mutant by valorphin is its more efficient hydrolysis.

The MD simulations showed that the effect of the R669A mutation on the binding of the synthetic Arg_2_-2NA substrate was much smaller than on the binding of peptides. For the WT–Arg_2_-2NA and R669A–Arg_2_-2NA complexes, similar binding modes (see Figure 5) were observed during simulations, as indicated by the equally stable binding to the lower domain β-sheet (Appendix A), the total number of hydrogen bonds between the protein and the ligand (Appendix A), and the similar MM/GBSA per-residue binding-free energies (Figure 2) in Arg_2_-2NA complexes with all protein variants. Although the total numbers of hydrogen bonds between proteins and ligands were almost the same in both complexes, some different amino acid residues were involved in stabilization, e.g., residues N391 and D496, which formed more hydrogen bonds with Arg_2_-2NA when bound with the WT enzyme compared to residues E316, D396, and E329 when Arg_2_-2NA was bound to R669A (Figure 3). MM/PBSA binding energies (Table 2) indicate slightly more favorable conditions for the binding of Arg_2_-2NA with the WT enzyme compared to those of the R669A mutant, in contrast to a slightly lower *K*_M_ value determined for the R669A–Arg_2_-2NA complex compared to the WT–Arg_2_-2NA complex (Table 3). However, it is known that the *K*_d_ value (which can be derived from the binding free energy) cannot be strictly equated with the *K*_M_; thus, the comparison between them can only be made on a qualitative basis. The Michaelis constant (*K*_M_ value) is always larger than or equal to the dissociation constant, as besides the affinity a ligand has for the binding site, it also incorporates the reaction rate for the chemical step.

The main difference between the Arg_2_-2NA–protein interactions observed in MD simulations of the WT and mutant complexes is the tighter binding of the carbonyl group of the cleavable peptide bond and the zinc ion in the mutant complex. This stronger interaction of the substrate with a catalytically important metal center and a larger deviation from the Bürgi–Dunitz angle [23] defining the direction of attack of the hydroxide ion, could explain the somewhat lower *K*_M_ and decrease in turnover (lower *k*_cat_), respectively, determined for the R669A–Arg_2_-2NA complex compared to the WT–Arg_2_-2NA complex (Figure 6 and Appendix A, Table 3). It is important to note that the QM/MM calculations [17] showed tighter binding with the catalytically important metal center in the Michaelis complex of the good Leu-enkephalin substrate bound with DPP III (2.45 Å) compared to the slow tynorphin (3.00 Å), suggesting a good relationship to the measured *k*_cat_ values.

Apart from a slightly larger average Bürgi–Dunitz angle observed during the simulations of the R669A–valorphin complex compared to the complex with the WT enzyme, which is also consistent with the more efficient hydrolysis speculated for this complex, no significant change in the average Zn-P1(O) distance or O_P1_-C_P1_-Ow angle was observed for the other complexes with the three peptides (Figure 6).

The inhibition constants were also determined for the R669M mutant. While the effects of the R669M and R669A mutations on the *K*_i_ of Leu-enkephalin were the same, the effect of the R669M mutation on the slow DPP III substrates was significantly (~3-fold) less than the effect of the R669A mutation. The increase in *K*_i_ values for all peptides with mutants R669A and R669M (Table 1) compared to the WT enzyme can be explained by the absence of a salt bridge or hydrogen bond interaction between the R669 guanidino group and the peptide C-terminus or the carbonyl group at the P3’ position, respectively. We can speculate that this might be sufficient to explain the 5-fold increases in *K*_i_ values for Leu-enkephalin with mutant enzymes. The even greater increase in *K*_i_ values obtained for complexes between slow peptide substrates and the mutant enzymes R669A (~30-fold) and R669M (~10-fold) compared with the WT enzyme, in addition to the absence of these interactions, is probably due to the replacement of the cation–π interactions (between tryptophan at the P3’ position and R669) in WT by weaker CH–π interactions in the mutants. Because the methionine side chain and the indole group of tryptophan can also interact via CH–π interactions, we expect a smaller effect of the R669M mutation than R669A on slow substrate binding, as obtained experimentally (Table 1).

### 2.2. Impact of R399A Mutation on Ligand Binding

R399 is a conserved amino acid that is a part of the 2nd conserved region (G385-L409) of DPP III [24]. MD simulations of the WT protein in complexes with Leu-enkephalin and hemorphin-4, i. e. good DPP III substrates, indicated the formation of van der Waals and occasionally polar interactions between R399 and the side chain of the tyrosine residue at the peptide P2 position (see Appendix A).

According to the MM/PBSA binding energies (Table 2), the effect of the R399A mutation is the highest for Leu-enkephalin, i.e., the affinity of Leu-enkephalin to bind to the mutant enzyme is significantly lower than for the enzyme WT. Almost unchanged binding affinities were determined for the complexes with Arg_2_-2NA and tynorphin, whereas valorphin showed slightly more favorable binding to the R399A mutant compared to WT.

The greater effect of the R399A mutation on Leu-enkephalin binding is a consequence of the disruption of hydrogen bonds between the first three amino acids at the N-terminus of Leu-enkephalin and protein residues E316, H568, and A388 (Figure 3 and Appendix A). The R399A mutant lacks important hydrogen bonds between the P1 subsite and conserved H568, the stabilizing van der Waals and polar interactions between R399 and tyrosine in the P2 position, present in the complex with the enzyme WT, are lost by the mutation, and the hydrogen bonds with residues N391 and E316 are reduced (see Figure 3). The salt bridge interactions between the ligand C-terminus (at the P3’ position) and R669 present in the protein WT are replaced by less efficient interactions with R572. R572 simultaneously forms hydrogen bonds with the backbone of the residue at the P2’ position (Figure 3 and Appendix A) in the mutant R399A. Consistent with all these changes in interactions, per-residue MM/GBSA binding decomposition analysis showed less favorable binding for the Leu-enkephalin residues at the P2, P2’, and P3’ positions with the mutant enzyme than with the WT enzyme (Figure 2). It appears that the R399A mutation in complex with Leu-enkephalin impairs binding of the entire molecule from its N- to C-termini (Figure 7).

As mentioned previously, the effect of this mutation on the other ligands is small. In addition to further stabilizing the binding of the P2 and P1’ residues of valorphin with the β-strand from the lower protein domain, the R399A mutation had no effect on the antiparallel binding of Arg_2_-2NA, tynorphin, and valorphin (see Figure 2 and Appendix A). In the R399A–tynorphin complex, a lack of hydrogen bonding between the side chain of the P1’ residue (tyrosine) and the side chain and carboxyl group of E512 (Figure 3) resulted in less favorable binding of the tynorphin P1’ residue determined through per-residue MM/GBSA decomposition analysis (Figure 2).

The per-residue MM/GBSA decomposition analysis revealed better stabilization of Arg_2_-2NA and valorphin P1 subsites in complex with mutant R399A compared to enzyme WT (Figure 2). In the complex with Arg_2_-2NA, this was the result of an additional salt bridge interaction between the guanidinium group of the residue at the P1 subsite of the ligand and Glu327 (Figure 3), together with the salt bridge interaction with E329 that was also present in the WT complex. In the complex with valorphin, the more favorable binding determined for the valine residue in the P1 position in the complex with mutant enzyme than with WT was mainly a consequence of the stronger hydrogen bonding with the conserved H568 and Y318 residues (Figure 3 and Appendix A), as also observed in the complex with mutant R669A.

Interestingly, the R399A mutation appeared to destabilize the hydrogen bonding between R669 and the residue at the P3’ position of the penta-peptide ligands (i.e., tynorphin and Leu-enkephalin) (Figure 3) but not the longer ligands (i.e., hepta-peptide valorphin). As mentioned previously, in the complex with Leu-enkephalin, this was compensated by strong hydrogen bonding with R572, whereas in the complex with tynorphin, there was occasional hydrogen bonding with V412 (Figure 3). Better stabilization of the P3’ subsite in complex with Leu-enkephalin than in complex with tynorphin was also indicated by the results of the per-residue MM/GBSA decomposition analysis (Figure 2). Interestingly, both R399A and R669A mutations resulted in partial opening of the enzyme when tynorphin was bound (Appendix A). The gentle opening of the enzyme binding site and the absence of the stronger interactions of tynorphin with the upper protein domain (e.g., hydrogen bonding with residues R572 or E512) caused the ligand to move further away from the metal ion (see Zn–P1(O) distance in Figure 6a) while continuing to be bound to the β-strand of the lower protein domain (Appendix A). By moving away from zinc, tynorphin simultaneously lost its hydrogen bond with the conserved H568 (Appendix A).

The results of the MD simulations suggest that the effect of the R399A mutation would be most pronounced in complex with Leu-enkephalin. This mutation not only impairs ligand binding by hindering antiparallel binding to the β-strand of the lower domain, which significantly increases the binding energy, but also disrupts hydrogen bonding to the catalytically important H568, which would consequently complicate the hydrolysis reaction. The R399A mutation appears to affect the binding and hydrolysis of tynorphin, as less favorable bindings are observed for the P1’ and P3’ subsites, and the P1 subsite moves away from the catalytic metal ion and hydrolytically important amino acids (such as H568 and Y318). On the other hand, the R399A mutation seems to contribute to a more stable binding of valorphin and a better possibility of hydrolysis, whereas Arg_2_-2NA binding and hydrolysis are probably least affected.

## 3. Materials and Methods

### 3.1. Experimental Methods

#### 3.1.1. Protein Expression and Purification

Human DPP3 wild-type (WT) and variants R669A and R669M were expressed in *E. coli* and purified using IMAC, as described in detail recently [18]. Briefly, variants were prepared using a QuikChange II site-directed mutagenesis kit (Agilent Technologies, Santa Clara, CA, USA) while following the manufacturer’s instructions, with the hDPP3 gene on a pET21a plasmid as template [25]. Complementary primer pairs were ordered from Macrogen Europe (The Netherlands) with the following sequences:

R669A_F GCTGCGTAAGGAATCTgcGAAGCTCATTGTTCAGCC;

R669A_R GGCTGAACAATGAGCTTCgcAGATTCCTTACGCAGC;

R669M_F GCTGCGTAAGGAATCTatGAAGCTCATTGTTCAGCC;

R669M_R GGCTGAACAATGAGCTTCatAGATTCCTTACGCAGC.

Proteins were expressed in *E. coli* for approximately 20 h at 18 °C, shaken at 120 rpm, and paired with 0.25 mM IPTG as an expression inductor. Cell pellets were lysed with lysosyme and with sonication; the lysate was then purified with DNAse I and through centrifugation, and affinity chromatography was performed on a Rotigarose His/Ni column. Lysis, wash, and elution buffers contained imidazole at concentrations of 10 mM, 20 mM, and 300 mM, respectively, and all buffers contained 50 mM Tris pH = 8.0, 300 mM NaCl. Fractions were pooled based on protein concentration (determined by using a BioDrop microvolume spectrophotometer) and purity (assessed by SDS-PAGE), and they were desalted on a PD-10 column with 20 mM Tris-Cl buffer, pH = 7.5. Aliquots were frozen and stored at −80 °C.

The R669A and R669M mutations were selected to distinguish the effects of the size of the arginine residue and the presence of the guanidine group. The mutation to alanine is the most common type of mutation in biochemical research because it constitutes the removal of the side chain, but methionine was chosen as a more conservative option.

#### 3.1.2. Enzyme Kinetics and Inhibition

Enzyme kinetics were measured using an Agilent Cary Eclipse fluorescence spectrophotometer (Agilent Technologies, Santa Clara, CA, USA) with Arg_2_-2NA as a substrate analog and fluorescent 2NA as the product of the peptidase reaction. The assay has been described in detail previously [16,18]. Arg_2_-2NA and peptide in 20 mM Tris-Cl buffer pH = 7.4 were pre-incubated at 25 °C for 2 min before the reaction was started by adding the enzyme into the cuvette. Substrate concentrations ranged from 0.25 to 100 μM. The peptide (inhibitor) concentrations were chosen so the output could be easily measured. For tynorphin, a range of 1–50 nM was used for the measurements with the wild-type enzyme and 1–5 μM for the mutants; for Leu-enkephalin, 5–30 μM for the wild-type and 180–360 μM for the mutants; for valorphin, 15–150 nM for the wild-type and 0.75–11.2 μM for the mutants. Three peptide concentrations were used for each peptide. The enzyme concentration was 1 nM in all measurements. After measuring Michaelis–Menten kinetics with the artificial substrate, Lineweaver–Burk plots were used to confirm that peptides acted as competitive inhibitors to artificial substrate. Kinetic parameters and inhibition constants were determined via nonlinear regression in GraphPad 5 (GraphPad Prism 5.0.1, San Diego, CA, USA) as best-fit values and standard errors, using a pre-set analysis for competitive inhibition.

### 3.2. Computational Methods

#### 3.2.1. System Preparation

The structure of the ligand-free hDPP III in closed form, deposited in the PDB under the code 5EGY, was used as a template structure for the simulations of the unbound WT enzyme. Prior to the simulations, all mutated residues present in this X-ray structure were converted back to the wild-type variant (S19C, C207E, A451E, C491S, S519C, and S654C). The R399A- and R669A-mutated enzyme structures were prepared by editing the structure of the wild-type enzyme. All of these mutations were performed using tleap, a basic preparation program for simulations from the AMBER20 program package (http://ambermd.org/, downloaded June 2020.).

The complex of the WT DPP III with the synthetic substrate Arg_2_-2NA was prepared based on the structure of the DPP III–tynorphin complex (PDB code: 3T6B), wherein the positions of the two arginine residues at the N-terminus of Arg_2_-2NA coincided with the positions of the two valine residues at the N-terminus of tynorphin. The WT complexes with Leu-enkephalin (YGGFL), tynorphin (VVYPW), and valorphin (VVYPWTQ) were prepared and simulated in our previous publication [18]. Mutated complexes with different ligands were prepared using the original wild-type structures as templates.

All Arg and Lys residues were positively charged, whereas the Glu and Asp residues were negative. With the exception of H568, which was positively charged, all other histidines were in a neutral state. Those coordinating the zinc ion, H450 and H455, were Nδ, and other were Nε protonated. Parameterization was performed within an ff19SB force field [26], and for the zinc ion, the extended 4-ligand hybrid bonded/non-bonded parameters derived in our previous work [27] were used. The complexes were neutralized with Na+ ions and solvated in an octahedral box filled with OPC–water molecules [28]. The minimum distance between the solvated complex and the edge of the box was 11 Å. The solvated systems were minimized, followed by heating, density equalization, and productive MD simulations, as described in our previous publication [18].

#### 3.2.2. MD Simulations

All simulations were carried out using the AMBER20 suite of programs [29].

System minimization was performed in 3 cycles, with 1500, 2500, and 1500 steps for the ligand-free protein structures and for DPP III complexes with Leu-enkephalin and tynorphin, whereas for the complexes with Arg_2_-2NA and valorphin, for which X-ray structures are not available, an additional minimization cycle (4500 steps) was conducted without any constrains. In the first cycle, water molecules were relaxed while the rest of the system was harmonically restrained with a force constant of 32 kcal mol^−1^ Å^−2^. In the second and third cycles, the protein and peptide backbone atoms were restrained with force constants of 12 kcal mol^−1^ Å^−2^ and 6 kcal mol^−1^ Å^−2^, respectively.

The energy-optimized systems were heated to 300 K (30 ps, *NVT* ensemble), and the density was equilibrated (970 ps, *NpT* ensemble). The equilibrated systems were simulated for 1 μs under *NpT* conditions. During heating and equilibration, the time step was 1 fs, and during the productive MD simulations, it was 2 fs. The SHAKE algorithm was used to constrain covalent bonds with hydrogen atoms. The pressure was maintained at 1 atm using the Berendsen barostat [30], and the system temperature was kept constant at 300 K using the Langevin thermostat [31]. Simulations were performed using periodic boundary conditions (PBC) with a cutoff of 11 Å, and the particle mesh Ewald (PME) method was used to calculate long-range electrostatic interactions [32,33]. Details of the MD simulations can be found in our previous publication [18]. To allow relaxation of the protein by binding a longer (hepta-peptide) ligand instead of a shorter one (penta-peptide), we restrained valorphin residues at the P2, P1, and P1’ positions and the protein residues E316, Y318, and H568 with the harmonic force constant of 32 kcal mol^−1^ Å^−2^ during heating, solvent density equilibration, and during 10 ns of MD simulations at 300 K of wild-type and mutant enzyme complexes’ MD simulations. Then, the systems were gradually relaxed in 4 series of 10 ns MD simulations at 300 K. In the first stage, the side chains of protein residues E316, Y318, and H568 were relaxed; in the second and third stages, the side chains of peptide residues P2, P1, and P1’ were also relaxed; in the fourth stage, only the backbone atoms of peptide residues P2, P1, and P1’ were restrained. These were followed by 1 μs of unconstrained MD simulations.

#### 3.2.3. MM/PBSA and Per-Residue MM/GBSA Calculations

The free energy of ligand binding was approximated by the energy calculated in the AMBER20 program MM/PBSA (Molecular Mechanics Poisson-Boltzmann Surface Area) [34]. The calculations were performed for the enzyme with dielectric constant 2.0 immersed in the solvent with dielectric constant 80. The ion concentration was 0.02 M. The polar component of the enthalpy of solvation was calculated by using the Poisson–Boltzmann method, and the nonpolar component was determined from changes in the surface area accessible to the solvent, as described in our previous publication [18]. To explain the differences in binding free energies and identify residues critical for ligand binding, the energies were decomposed into the contribution of each residue using the Molecular Mechanical Generalized Born Surface Area (MM/GBSA) approach implemented in the AMBER20 software. Calculations were performed for the enzyme with a relative permittivity of 1.0 in a solvent with a relative permittivity of 80.0.

Both MM/PBSA and per-residue MM/GBSA calculations were performed for the last 0.8 µs of the MD simulations (from 200th to 1000th ns) while sampling the structures every 100 ps and using a single trajectory approximation.

We calculated the contribution of enthalpy to the binding free energy because conformational entropy is usually neglected due to its high computational cost when only the relative binding free energies of similar ligands are considered.

#### 3.2.4. Analysis

The results of the MD simulations were analyzed using the cpptraj module of Amber20. Boxplot diagrams of the results (distances and angles) of the cpptraj analysis were generated using the Origin 7.0 program (OriginLab Corporation, Northampton, MA, USA). Origin’s boxplots represent summary statistics, as the box is defined by the 25th and 75th percentiles, a vertical line that goes through a box is its median, the small square is the average value, and the whiskers represent the standard deviation.

## 4. Conclusions

Using a combined experimental and computational approach, we determined the effects of mutating the conserved arginines at positions 399 and 669 on the binding of various DPP III substrates: synthetic substrate Arg_2_-2NA and peptide substrates Leu-enkephalin, tynorphin, and valorphin.

MD simulations show that mutations R399A and R669A have not altered the ligand-free protein’s overall structure and dynamic. This is also true for the complexes with the good substrates (Arg_2_-2NA and Leu-enkephalin) and valorphin, whereas in the mutated complex with tynorphin, a moderate protein opening was observed compared to the WT enzyme.

Mutation of R669 to Ala or Met was found to significantly impair binding of the slow substrates tynorphin and valorphin, whereas the effects on binding of the good substrates Arg_2_-2NA and Leu-enkephalin were small. However, although the R669A(M) mutation decreased the stability of the protein complex with tynorphin, no major change in the stability of the mutated DPP III–valorphin complex compared with the complex with the WT enzyme was observed during MD simulations. Most importantly, analysis of the simulation results showed that the residues defining the cleavable peptide bond (i.e., the residues at positions P1 and P1’) were bound with higher affinity and with higher capacity for peptide hydrolysis in the binding site of R669A than in the binding site of the enzyme WT. Therefore, the lower potency (higher *K_i_*) of tynorphin and valorphin to inhibit hydrolysis of Arg_2_-2NA in the mutant R669A than in the WT enzyme can be explained by the lower binding affinity of the former and probably the higher ability of the latter to be hydrolyzed in the mutant than in the enzyme WT.

The experimentally determined effect of the R669A mutation on the potency of Leu-enkephalin to inhibit hydrolysis of the preferred synthetic substrate Arg_2_-2NA was a combination of two effects determined by the higher ability of Arg_2_-2NA to be hydrolyzed in the mutant than in the enzyme WT and the lower binding affinity of Leu-enkephalin for the mutant than for WT.

From the results of MD simulations, we can conclude that the effect of the R399A mutation is most pronounced in complex with Leu-enkephalin. This mutation not only impairs ligand binding by hindering antiparallel binding to the β-strand of the lower domain, which significantly increases binding energy, but also disrupts hydrogen bonding to the catalytically important H568, which would consequently complicate the hydrolysis reaction. The R399A mutation also appears to affect the binding and/or hydrolysis of slow substrates, whereas Arg_2_-2NA binding and hydrolysis are probably the least affected.

## Figures and Tables

**Figure 1 molecules-28-01976-f001:**
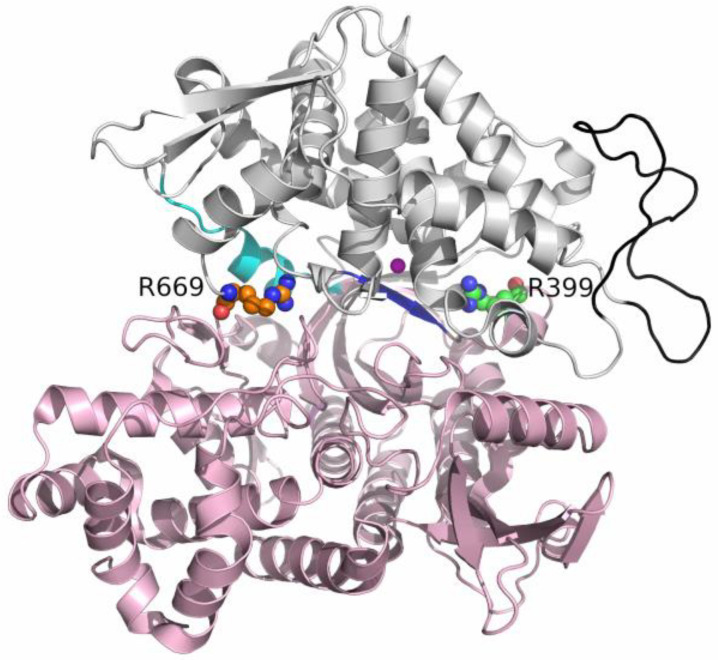
Location of the mutated arginine residues in the unbound enzyme structure from the PDB with code 5EGY. The lower protein domain residues (1–336, 375–420, and 669–726) are colored light pink, and the upper domain residues (337–374 and 421–668) are in gray. The flexible loop (residues 463–489) is colored black. The hinge (residues 409–420) involved in the conformational change of the protein is cyan, and the β-strand to which the ligand binds is colored blue. The zinc ion is colored magenta. Arginines 669 and 399 are in orange and green, with N and O atoms in red and blue.

**Figure 2 molecules-28-01976-f002:**
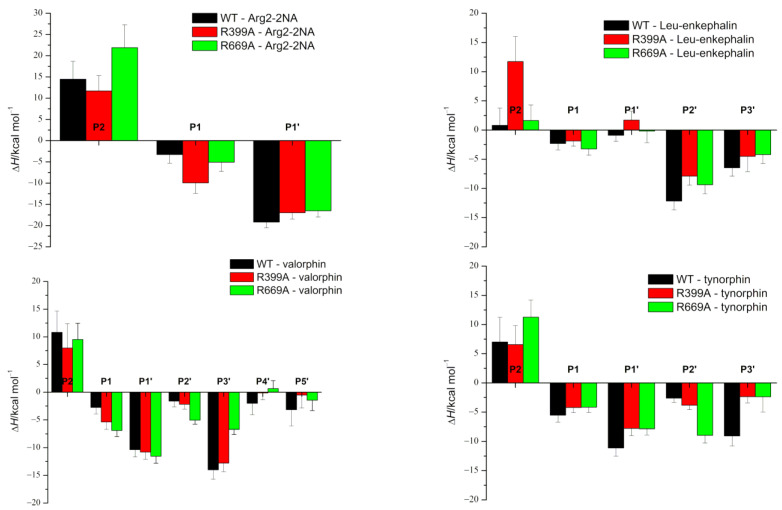
The MM/GBSA per-residue binding-free energies indicate how much each ligand residue contributes to the binding-free energy. Calculations were performed using the set of conformers sampled during the last 800 ns of the MD simulations of the complexes. The amino acid residues of the peptide ligand are denoted as P1 to Pn and P1’ to Pn’, counting from the scissile peptide bond to the N- and C-termini of the peptides, respectively.

**Figure 3 molecules-28-01976-f003:**
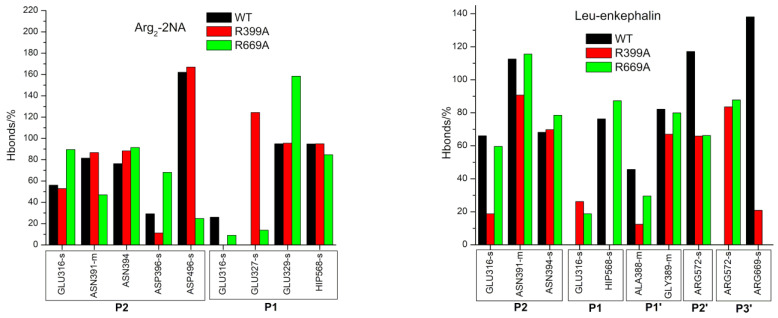
Hydrogen bond occupancy (shown only when occupancy > 20% in at least one complex) calculated by using the Hbonds plugin (VMD) between the ligand and the rest of the protein. Angle and distance cut-offs were 45° and 3 Å, respectively. Indicated amino acids participated as hydrogen bond donor and/or acceptors (for more details see Appendix A). When applicable, it is also indicated whether only the atoms of the side chain (s) or of the main chain (m) were involved in hydrogen bonding.

**Figure 4 molecules-28-01976-f004:**
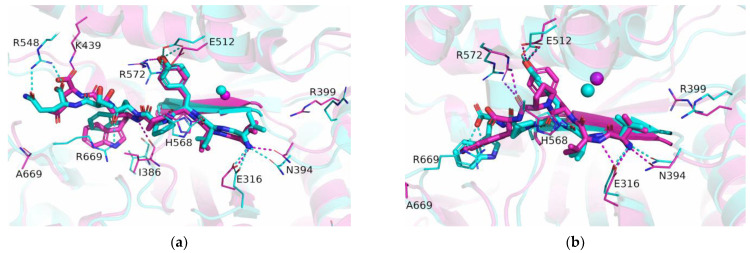
Binding of valorphin (**a**) and tynorphin (**b**) with the wild-type (cyan) and the R669A mutated (magenta) DPP III enzymes. Shown are optimized structures obtained after 1 µs of MD simulations. The amino acid residues that formed hydrogen bonds (indicated by dashed lines) with the ligand are shown as thin sticks, as are the amino acid residues that formed polar interactions with the ligand. Mutated residues are also shown as thin sticks. The β-sheet (residues A388-N391) from the lower protein domain involved in antiparallel binding of the ligand is shown as an opaque cartoon. The zinc ion is shown as a sphere. The hydrogen atoms are not shown, nor are the main chain atoms, except at A699.

**Figure 5 molecules-28-01976-f005:**
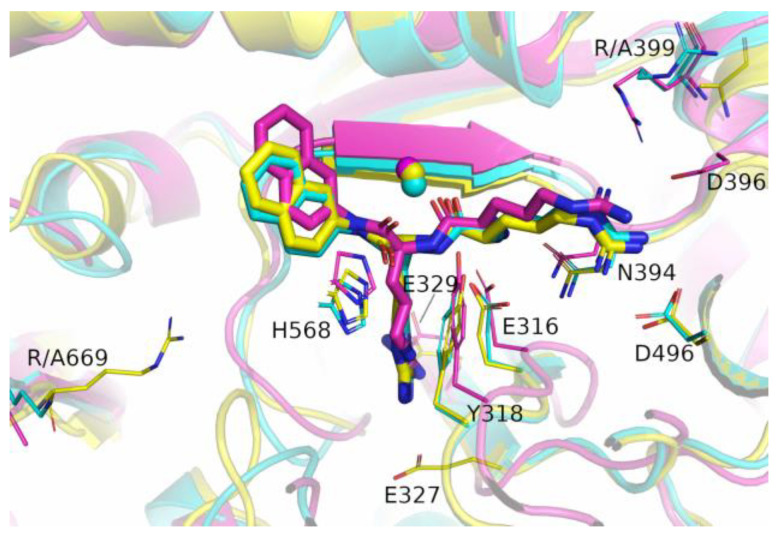
Binding of Arg_2_-2NA in the binding site of the wild-type (cyan) and the mutated (R399A in yellow and R669A in magenta) DPP III enzymes. Shown are optimized structures obtained after 1 µs of MD simulations. The mutated amino acids and the amino acid residues that form polar interactions/hydrogen bonds with the ligand are shown as thin sticks. The β-sheet (residues A388-N391) from the lower protein domain involved in antiparallel binding of the ligand is shown as an opaque cartoon. The zinc ion is shown as a sphere.

**Figure 6 molecules-28-01976-f006:**
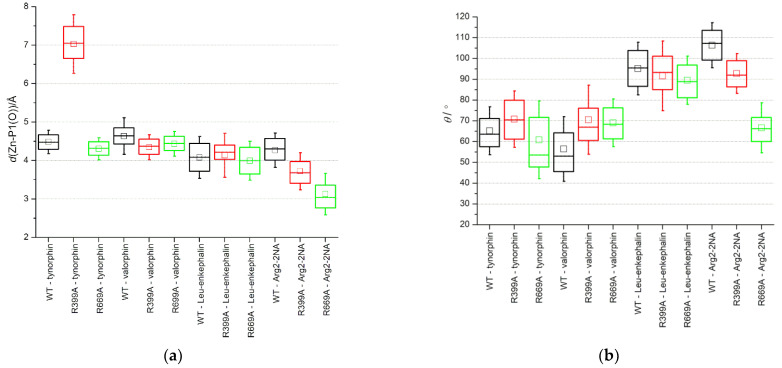
Box plots of (**a**) the distance between the zinc ion and the peptide carbonyl oxygen atom (O) at the P1 position and (**b**) the angle defining the direction of OH^−^ attack calculated between the carbonyl oxygen (O) and carbon (C) of the residue in position P1 and the oxygen atom of the activated water molecule (Ow). The activated water molecule simultaneously coordinated Zn and was hydrogen bonded to E451. The whiskers depict one standard deviation.

**Figure 7 molecules-28-01976-f007:**
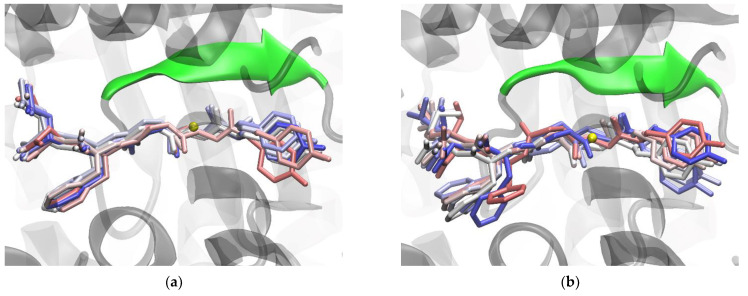
Binding of Leu-enkephalin with the wild-type (**a**) and the R399A mutant (**b**) DPP III - structures obtained after 1 µs of MD simulations. The Leu-enkephalin structures sampled every 100 ns (starting from 200th ns) are shown as sticks and colored according to their position in the trajectory (in light red at the 200th ns, in white in the middle of the trajectory, and in blue at the end). The β-sheet (residues A388-N391) from the lower protein domain involved in antiparallel binding of the ligand is shown as an opaque cartoon and is colored green. The zinc ion is shown as a yellow sphere. The hydrogen atoms are not shown.

**Table 1 molecules-28-01976-t001:** Inhibition constants of Leu-enkephalin, tynorphin, and valorphin determined for the DPP III catalyzed hydrolysis of Arg_2_-2NA.

	Wild-Type	R669A	R669M
*K*_i_ (tynorphin)/nM	11.2 ± 0.8	328 ± 25	102 ± 16
*K*_i_ (valorphin)/nM	36.5 ± 2.9	1207 ± 224	406 ± 33
*K*_i_ (Leu-enkephalin)/μM	10.4 ± 1.4	52 ± 5.3	58 ± 7.0

**Table 2 molecules-28-01976-t002:** The ligand binding enthalpy calculated using the MM/PBSA approach for the last 800 ns of MD simulations (for structures sampled every 100 ps).

	(Δ*H*_MM/PBSA_ ± SD)/kcal mol^−1^			
DPP III	Arg_2_-2NA	Leu-Enkephalin [YGGFL]	Valorphin [VVYPWTQ]	Tynorphin [VVYPW]
WT	−32.13 ± 7.36	−20.04 ± 4.52	−9.15 ± 7.01	−21.58 ± 4.62
R399A	−35.09 ± 7.96	0.00 ± 7.91	−15.33 ± 7.11	−18.82 ± 4.59
R669A	−22.74 ± 6.25	−13.67 ± 4.98	−16.30 ± 6.05	−7.32 ± 4.39

**Table 3 molecules-28-01976-t003:** Results of the Michaelis–Menten test for the hydrolysis of Arg_2_-2NA in the presence of DPP III variants.

	Wild-Type	R669A	R669M
*K*_M_/μM	7.8 ± 0.6	3.4 ± 0.3	2.8 ± 0.2
*k*_cat_/s^−1^	3.74 ± 0.07	2.24 ± 0.05	2.26 ± 0.03
*k*_cat_/*K*_M_ (M∙s)^−1^	4.8 × 10^4^	6.6 × 10^4^	8.07 × 10^4^

## Data Availability

Not applicable.

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
