# Peer review of "Influence of Mutations of Conserved Arginines on Neuropeptide Binding in the DPP III Active Site"

_molecules, 2023, doi:10.3390/molecules28041976_

Round 1

Reviewer 1 Report

In this work, Tomic et al describe the influence of the mutations of R669A and R399A in the catalytic activity of DPP III. Their findings indicate that these mutations affect the binding of target peptide substrates and, therefore, the KM. Their results are supported by experimental and computational data. Overall, the experiments and methods are well-designed, elaborated, and detailed. Especially, the computational work is really strong, but also easy to read and follow even for those that are not experts in the subject.

Author Response

Thank you!

Reviewer 2 Report

Very clear and well-written manuscript.

Revisions required are the following:

line 547: "alter" to "altered"

Author Response

Thank you for this correction: line 547: "alter"  was changed to "altered"

Reviewer 3 Report

The results and their value in the presented form are suitable for publication after supplementing with the following remarks.

Minor comments:

1. Please add more details to "3.1.2. Enzyme kinetics and inhibition". I understand that the citations were placed but it will be more convenient to analyze when the technical details of the determination of the inhibition constants are given.

2. The authors performed the mutation of position R699 to Ala or Met to show the role of arginine in the inhibition properties of slow substrates tynorphin and valorphin and the good substrates Arg2-2NA and Leu-enkephalin.

Why did the authors choose those amino acids residue? Do naturally occur that mutations? Please comment.  

Author Response

1. The revised manuscript contains more details on "3.1.2. enzyme kinetics and inhibition" as requested  in the review. 

2. The R669A and R669M mutations were selected to distinguish the effects of the size of the arginine residue and the presence of the guanidine group. The mutation to alanine is the most common type of mutation in biochemical research because it constitutes the removal of the side chain, while methionine was chosen as a more conservative option.

This comment was added at the end of paragraph 3.1.1